# Estimating COVID-19 mortality in Italy early in the COVID-19 pandemic

Chirag Modi[1]✉, Vanessa Böhm 🔟 [1,2,4], Simone Ferraro[1,2,4], George Stein 🔟 [1,2] & Uroš Seljak[1,2,3]

Estimating rates of COVID-19 infection and associated mortality is challenging due to uncertainties in case ascertainment. We perform a counterfactual time series analysis on overall mortality data from towns in Italy, comparing the population mortality in 2020 with previous years, to estimate mortality from COVID-19. We find that the number of COVID-19 deaths in Italy in 2020 until September 9 was 59,000–62,000, compared to the official number of 36,000. The proportion of the population that died was 0.29% in the most affected region, Lombardia, and 0.57% in the most affected province, Bergamo. Combining reported test positive rates from Italy with estimates of infection fatality rates from the Diamond Princess cruise ship, we estimate the infection rate as 29% (95% confidence interval 15–52%) in Lombardy, and 72% (95% confidence interval 36–100%) in Bergamo.

[1] Berkeley Center for Cosmological Physics, Department of Physics, University of California, Berkeley, CA, USA. [2] Lawrence Berkeley National Laboratory, One Cyclotron Road, Berkeley, CA, USA. [3] Berkeley Institute for Data Science, University of California, Berkeley, CA, USA. [4] These authors contributed equally: Vanessa Böhm, Simone Ferraro. ✉email: cmodi@flatironinstitute.org

The COVID-19 pandemic is one of the most pressing challenges the world is facing today. Despite a large number of infected individuals and confirmed deaths, large uncertainties about the properties of the virus and the infection still remain. In this article we present an analysis of the mortality rate in Italy in 2020, which we find has been significantly higher than in previous years. Italy was one of the hardest-hit countries in the early stages of the pandemic with >400,000 confirmed cases and >36,000 COVID-attributed deaths as of mid-October 2020[1].

Several numbers in Italy present statistical peculiarities such as the case fatality rate (CFR, defined as the ratio between the number of deaths attributed to COVID-19 and the number of positive tests), which is still at 10% in October 2020 and was much higher in the earlier stages of the pandemic[2], and has led to early estimates of high mortality[3]. The CFR is heavily affected by issues unrelated to the underlying disease, such as the extent of testing. A better metric is the infection fatality rate (IFR, the ratio between the number of deaths and the total number of infections), the knowledge of which is paramount to guide the public health response. The IFR, along with the population fatality rate (PFR, defined as the ratio between the number of deaths and the total population), allows us to estimate the infection rate (IR, the fraction of the population that is infected), which estimates how wide-spread the diseases is in the population and which informs government response.

Estimating IFR and IR is challenging, both owing to limited testing (hence, poorly known number of infections) and the uncertainty in the number of fatalities attributed to COVID-19. Official data account for those that have been tested. However, there may have been other deaths that were not tested and went unrecorded, which would imply an underestimate of the death rate by the official COVID-19 numbers.

Given the uncertainties in the official COVID-19 fatality rate, it is important to explore other paths for obtaining it. In this article, we propose a counterfactual analysis: we use historical mortality rate data from Italy to construct models for the expected death rate in 2020 in the absence of the COVID-19 pandemic. We attribute the difference between the observed mortality in 2020 and the predicted counterfactual to the COVID-19 pandemic. The models we employ account for the historical year-to-year variability owing to seasonal effects such as the flu. We use two different models, each based on different assumptions about the underlying data distribution. One model is based on a conditional Gaussian Process (hereafter, referred to as CGP model) and the other on a Synthetic Controls Method (hereafter, SCM). Consistent results from these approaches suggest that the results are independent of the model assumptions.

## Results

Figure 1 shows the counterfactual predictions for all regions in Italy in 2020. We plot predictions from both, SCM (yellow) and CGP (green). For comparison, we also show the historical 2015–2019 data and their mean (gray), as well as the mortality in 2020 after accounting for the reported COVID-19 deaths (black), i.e., the total reported mortality minus the reported COVID death count. We note that the SCM and CGP methods both trace the pre-pandemic data closely (the latter method is designed to match the pre-pandemic data exactly as detailed in the Methods section) while the historical mean estimates are generally higher.

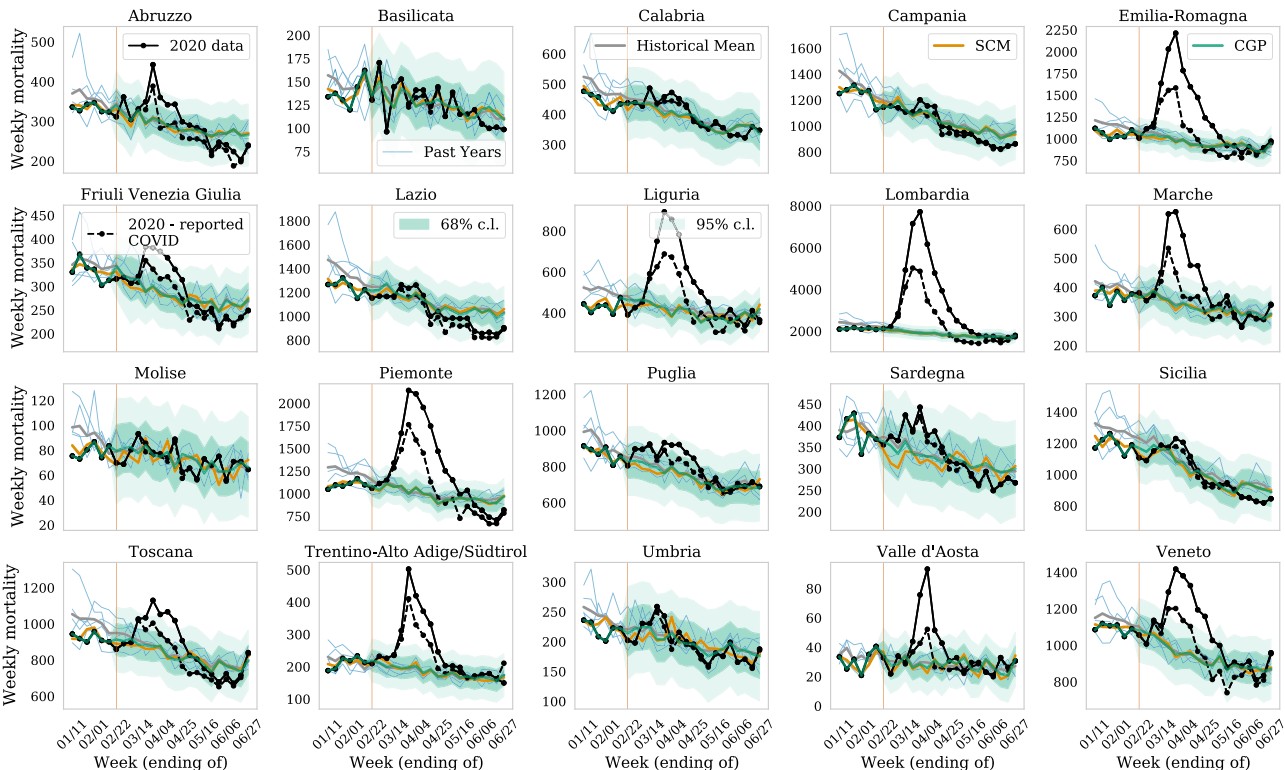

**Fig. 1 Validating counterfactuals for the pre-pandemic data.** we show the observed weekly mortality due to all causes for the period of 1 January to June 27 (black) in all 20 regions in Italy, and our prediction for the expected mortality in the absence of COVID-19 (conditional Gaussian Process (CGP) with its 1 and 2−σ error from the variance of the Gaussian model, i.e., 68% and 95% confidence interval, respectively, in green and synthetic controls method (SCM) in orange). The first reported COVID-19 mortality occurred in the week ending on February 22 (thin red vertical line). The historical data from 2015 to 2019 (blue) and corresponding historical mean (gray) is shown for comparison and are not a good fit to the observed pre-pandemic data. In the dashed-black line, we also show the observed mortality after removing reported COVID-19 deaths.

The mortality in Italy has been below-average in the first 2 months of 2020, probably owing to a milder than usual flu season. This discrepancy demonstrates why SCM and CGP provide better counterfactuals than the simple historical mean estimate: they take into account this year's pre-pandemic mortality and exploit the time-correlations in the mortality rates allowing to make more accurate and precise predictions, forgiven the different assumptions made by these methods hold true. Where our predicted excess mortality is lower than the reported COVID-19 fatalities, we will use the latter for our estimate of COVID-19 deaths. This selection only makes a statistically significant difference for the region of Lazio and age groups below 30 years of age in few other regions, but otherwise does not affect our conclusions with any significance. Figure 1 shows a clear excess in mortality over the counterfactual predictions after the week ending on 22 February, when the first COVID-19-related deaths were reported in Italy. This excess is primarily seen in the Northern regions, which are the hardest hit. In the remainder of this work, we focus on these regions and the province of Bergamo (see also ref. [4] for an earlier analysis).

In Fig. 2, we show the excess deaths over the expected counterfactual for every week of reported data. We focus on the few regions that were hardest hit by the pandemic and which lead to the most statistically significant conclusions. Figure 2a shows that the excess weekly mortality is significantly higher than the official COVID-19 deaths in all regions, as the beginning of the pandemic. We only have access to reported COVID-19 deaths in Bergamo up to 1 May 2020 (shown in a dashed pink vertical line) and beyond that, we extrapolate it in the same proportion as Lombardia, the region that the province of Bergamo resides in. Since May 2020, the estimated excess is less than reported COVID-19 fatalities for some regions, though it is mostly still

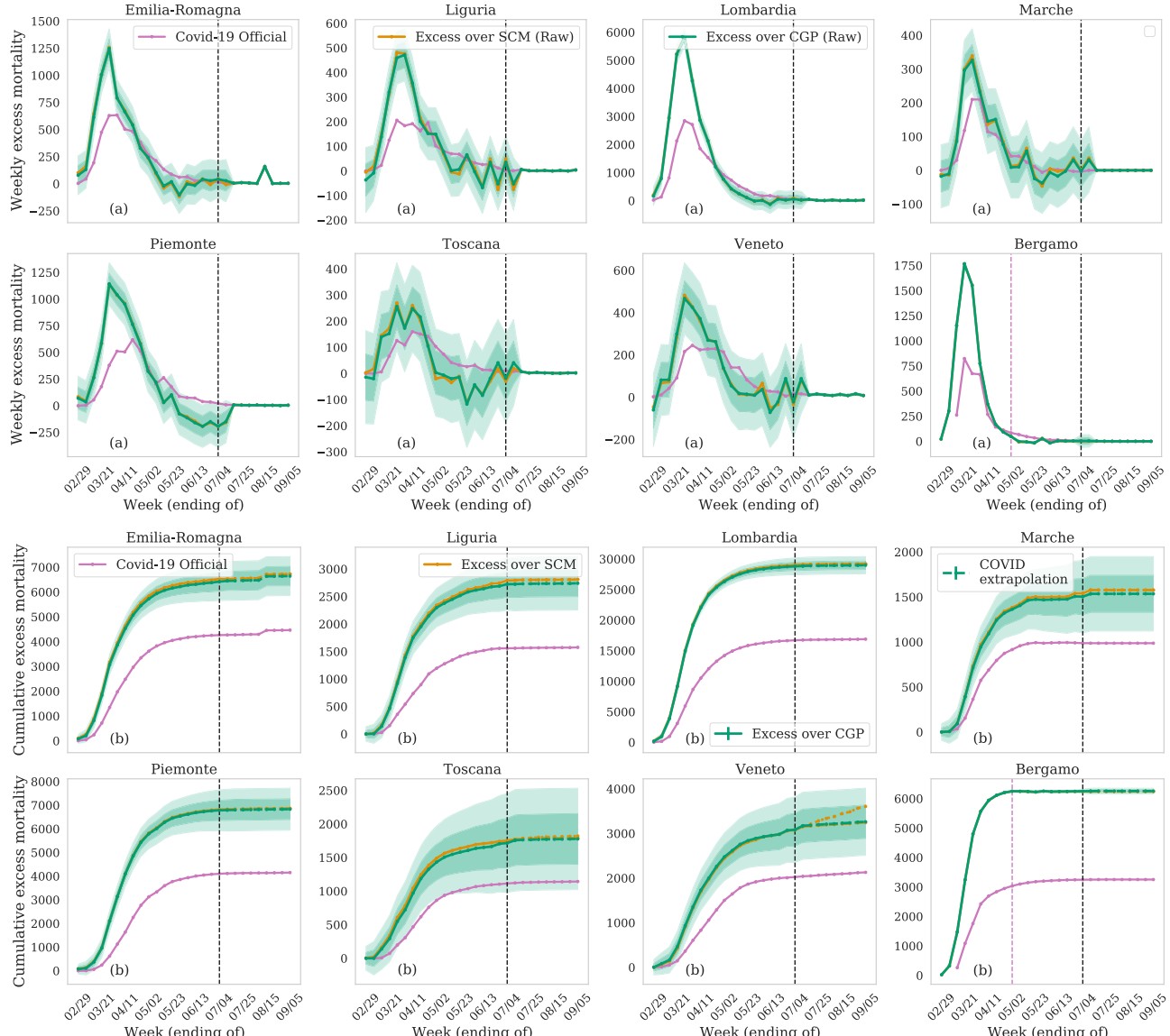

**Fig. 2 Excess mortality compared with reported COVID-19 deaths in regions of Northern Italy and the province of Bergamo.** (**a**) Excess weekly deaths, and (**b**) cumulative excess deaths, over the predicted counterfactual in comparison to the reported COVID-19 deaths (in pink) for the period since February 23rd (available COVID-19 data). Estimates from both the synthetic controls method (SCM, orange) and conditional Gaussian Process (CGP, green) counterfactuals agree. We show 1 and 2−σ error (68% and 95% confidence interval) from the variance of the Gaussian model. We find that COVID-19 deaths are under-reported by multiple factors for every period and every region. We extrapolate the data excess beyond June 27th, which is the last week with available total mortality data (dashed-black line), with dashed-lines. To do this, we make the conservative assumption after June 27 that the reported COVID-19 deaths are accurate and account for the excess mortality over predicted trends.

consistent with the $1-\sigma$ (68%) confidence interval of our predictions (except in Piemonte and Toscana). This is to be expected, as we move away from the intervention (the start of the pandemic), which happened in February, the counterfactual prediction becomes less accurate and will be more heavily influenced by the global mean. For these weeks and regions, whenever our predicted excess is less than the reported COVID-19 deaths, we will use the latter for estimating fatality rates and infection ratios (IRs). Figure 2b shows the cumulative excess in mortality compared with the total reported COVID-19 deaths at the end of each week. All regions see a consistent rise in excess deaths until early May 2020. If we attribute these deaths to COVID-19 infections, this implies that the worst affected regions such as Lombardia and Emilia-Romagna have likely underestimated the mortality by factors of 1.5, whereas other regions like Piemonte and Toscana have underestimated mortality by a factor of 2. For most regions, the number of deaths has decreased significantly since May 2020.

In Figs. 2 and 3, we have extrapolated our estimated excess from 27 June 5 September. Since the number of estimated excess deaths is consistent with reported COVID-19 deaths for the last 8 weeks for all regions (except Piemonte and Toscana), we assume that the weekly excess mortality is the same as the reported COVID-19 deaths for extrapolation after 27 June. Based on this, we estimate that the number of COVID-19 deaths in Italy is between $59{,}000-62{,}000$ as of 9 September 2020, more than a factor of $1.5\times$ higher than the official number. In the remainder of the paper, we will use this extrapolation to estimate the age-dependent and population fatality rates and IRs.

In Fig. 3, we show the excess mortality for different age groups in intervals of 10 years above the age of 40. We find some agreement between the estimated excess and the reported

COVID-19 deaths below the age of 70, but observe a significant and increasing discrepancy for higher age groups. This seems to suggest that testing and consequently probably also treatment has been more complete for lower age groups.

**Attributing excess deaths to COVID-19**. To back the assumption that excess deaths are a consequence of the pandemic, we establish a correlation between the daily excess deaths over the counterfactual and the official COVID-19 deaths by means of regression analysis: we perform a two-parameter fit to the excess deaths by allowing the official deaths to be scaled and shifted. We infer the time-lag and amplitude of this fit by minimizing $\chi^2$. We find that best fits are obtained for time-lags of $-6$ days for Lombardia, $-7$ days for Emilia-Romagna, $-8$ days for Piemonte, and $-6$ days for Marche. The inferred amplitudes range between 1.2 and 1.6. We provide figures and details for this analysis in the supplemental material. Given that both data sets report the day of death not the day of report, the inferred time-lags suggest that the official COVID-19 mortality lags behind the total mortality. One possible reasons for this could be that hospital treatment postpones death on average by several days. A ramping up of testing with time could also cause this behavior.

However, correlation is not causation and attributing the excess death rate to COVID-19 is still a strong assumption. Hence, we discuss possible caveats. COVID-19 has put enormous pressure on Italy's medical system and social services. This could have led to fatalities that could otherwise be averted, causing us to overestimate the COVID-19 deaths. However, the pressure on the medical system is regional and likely sustainable for regions with a low number of official COVID-19 deaths, like Piemonte and Liguria. Instead, we consistently find a similar and very large

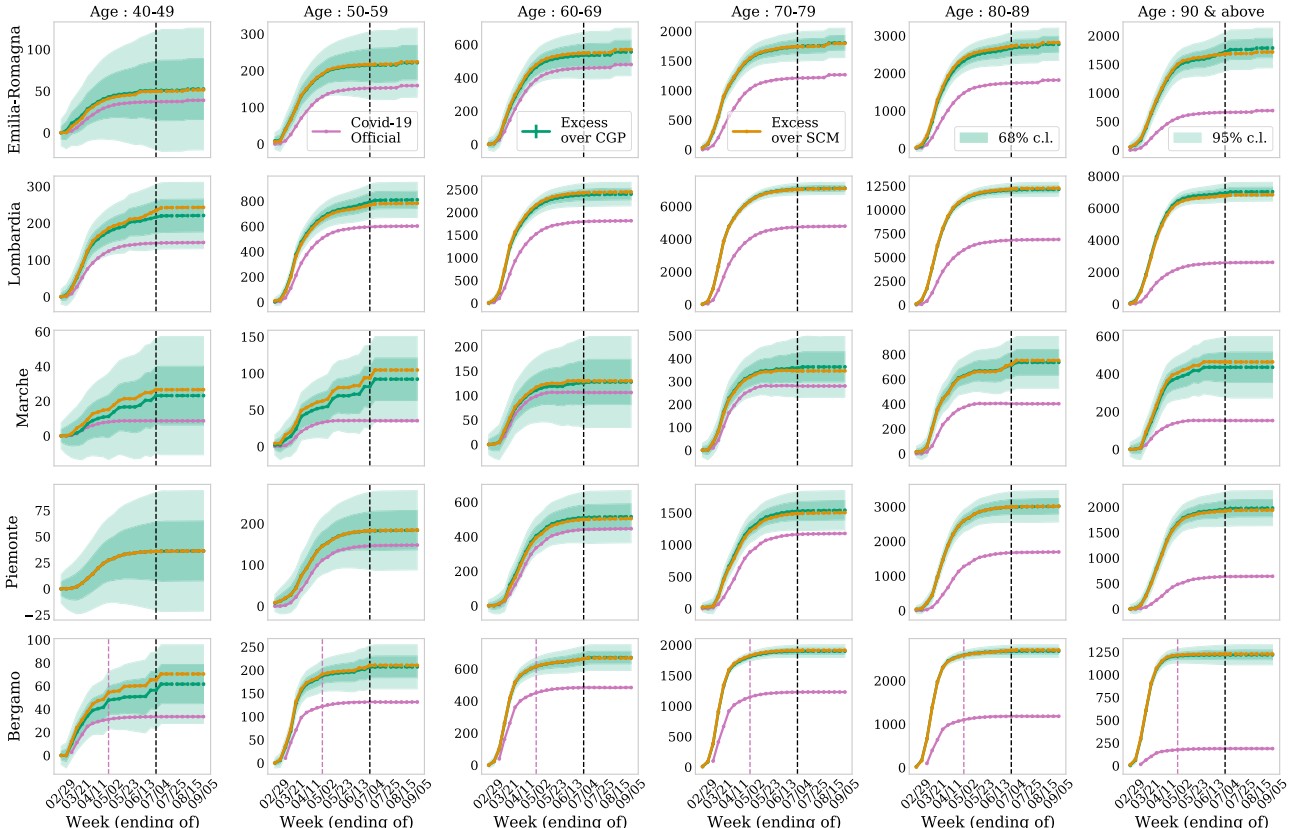

**Fig. 3 Age distribution of excess mortalities.** Same as Fig. 2b but for different age groups. We find a statistically significant excess over the reported COVID-19 deaths that is increasing with age.

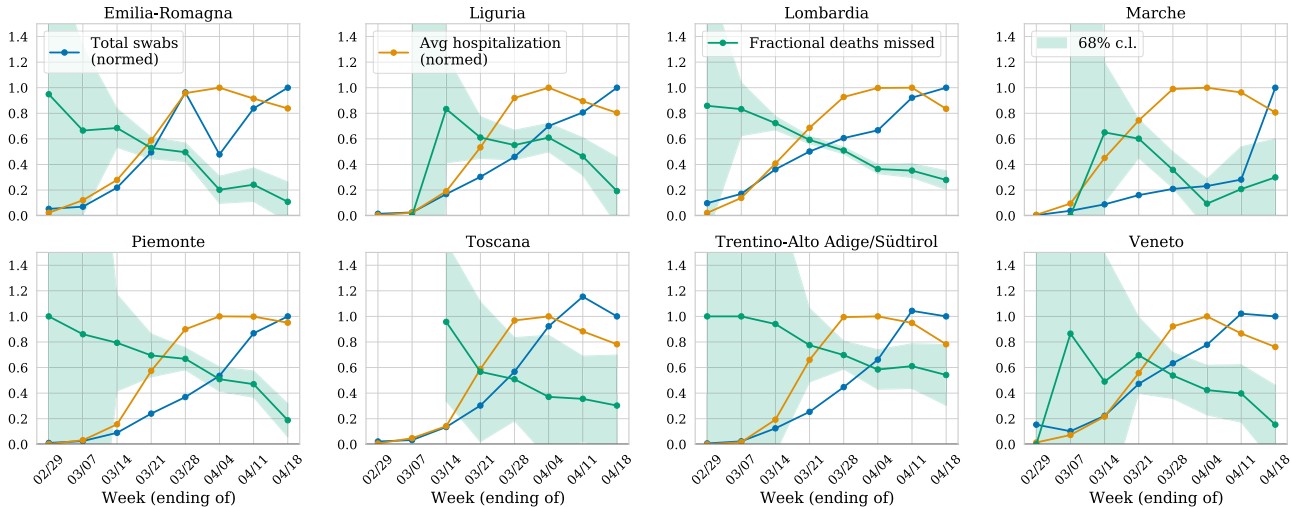

**Fig. 4 Fraction of missed deaths over time.** For the period of the pandemic, we show per week the fraction of missed deaths (green) with corresponding $1-\sigma$ (68% CI) estimated from the variance of the Gaussian model, the number of hospitalizations (normalized with the maximum weekly hospitalization up to 18 April 2020, in orange) and the number of tests conducted (normalized as a fraction of tests conducted in the week of 11–18 April 2020, in blue). We find that the missed fraction goes down as the number of tests increases while the hospitalizations have remained consistently high in the last 4 weeks.

excess in mortality over the official counts in many regions in Italy, which appears to be independent of how hard the region was hit (see Fig. 2b).

The temporal trend also lends a similar argument: the societal and medical systems should function normally in the earliest stages of the pandemic and get increasingly stressed as the number of infections increases. We see that the fraction of deaths missed by the reported COVID-19 fatalities is the highest in the early stage, and decreases as the number of reported infections increases. The reported COVID-19 fatalities finally catch up with the estimated excess fatalities by the end of April 2020. We show this in Fig. 4 where we compare the fraction of deaths missed every week with the number of COVID-19 reported hospitalizations (normalized with the maximum number of hospitalizations up to 18 April 2020).

Our hypothesis is that the excess deaths over official COVID-19 deaths are primarily due to the lack of testing in the initial stages of the pandemic. In Fig. 4, we also show the number of tests conducted every week as the fraction of tests conducted in the week of 11 April 2020. The trend supports our assertion that with an increase in testing as the pandemic evolves, the reported fatalities due to COVID-19 slowly catch up with the true current mortality and the increased pressure on medical systems did not have a statistically significant effect on the mortality.

There are also arguments that suggest we may have under-estimated the COVID-19 death rate. Italy has been under lockdown since 9 March 2020, which may have reduced fatalities due to other common causes such as road and workplace accidents, or criminal activities. This can be studied by observing the death rate correlations with the lockdown data in regions with little or no infection, such as South Italy. There are several regions that do not show an excess death rate, but none of them show a deficit death rate post 9 March 2020, so we assume that this effect is negligible, especially for age groups above working age.

**Fatality and IRs**. Having established that the observed excess deaths can reasonably be attributed to COVID-19, we can use our estimates and uncertainties of the excess mortality from the CGP and SCM counterfactuals to calculate the fatality rates and infection fractions for Italian regions. The left panel of Fig. 5

shows the PFR in different age groups, the total number of excess mortality deaths attributable to COVID-19 as a fraction of the population. We find a steep age dependence of PFR: in Bergamo province, 1.89%, 4.84%, and 11.06% of the entire population in the age groups 70–79, 80–89, and 90+, respectively, died. For the entire population, the PFR is 0.57% (and similarly 0.29% in Lombardia). Since the PFR corresponds to the IFR if the infection fraction is 1 (maximum possible), we expect these numbers to be the most conservative lower limits on the (age dependent) IFR (Table 2).

*Lower limits on IFR*. The central panel of Fig. 5 shows the lower bounds on the IFR. Estimating the IFR from the PFR requires the IR of the population. Here, we use the test positivity rate (TPR)—the fraction of positive to total tests, as an estimate of the fraction of the infected population. Owing to a lack of testing and the criterion of primarily testing people with symptoms, this should be an upper bound on the IR in the early stages of the pandemic. For every region, we use the maximum of the cumulative TPR estimated up to 5 September as our estimate for the IR. This should be an upper limit on the IR and hence give a conservative lower bound on the IFR. We further assume that this ratio is age-independent in every region[5]. The age-averaged lower bounds on the IFR are shown in Table 1, with the most robust estimate of $0.73 \pm 0.08\%$ IFR lower bound from Lombardia, consistent with 0.57% lower bound from Bergamo province.

*IR and IFR calibrated on the Diamond Princess*. The PFR can also be combined with an independent estimate of the IFR to obtain the IR with the relation IR=PFR/IFR. At the time of writing, the only large dataset with complete testing and hence unbiased estimate of the IFR is the Diamond Princess (DP) cruise ship. For our analysis, we assume that the age-dependent IFR is location independent: we account for age differences, but not for other differences between the DP and Italian populations in the same age group such as co-morbities, dose differences, or health-care access.

The last death on the ship was reported on 18 April 2020 and 11 out of 330 DP infections in the age group above 70 had been fatal (a few of the fatalities do not have age information). This results in an IFR for this age group of 3.3% and we assume a

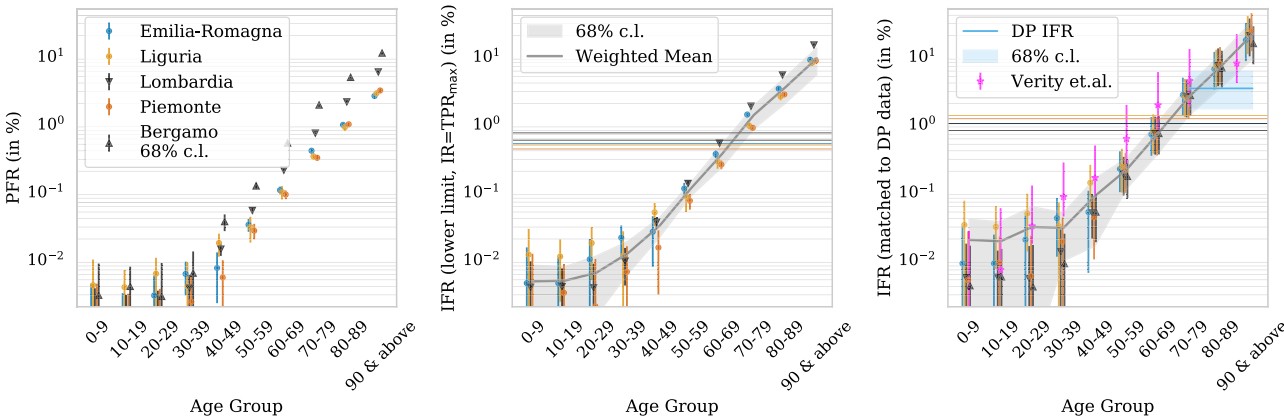

**Fig. 5 Fatality rates for different age groups and regions.** (Left) Population fatality rate (PFR) from the cumulative estimates divided by the regional population. (Center) Lower bounds on infection fatality rate (IFR) using the maximum test positive rate (TPR) as an upper bound on infection fraction. (Right) Estimates of the true IFR when normalizing the age 70–89 group to the Diamond Princess IFR (in shaded blue, with the corresponding Poisson error estimate). We also show estimates from Verity et al.[5] with corresponding (68% CI). in magenta, which gives less steep age dependence. In the center and right panel, the gray lines are weighted mean estimates for IFR with 1 sigma weighted standard deviation bands. The horizontal lines are the age-averaged IFR for the entire population. Error bars for all the regions and age groups are 1−$\sigma$ (68% CI) error from the variance of the Gaussian model combined with Poisson errors based on number of deaths that differs for every region and age group. In all panels, we have staggered the points horizontally for every age group for better visibility.

**Table 1 Estimated fatalities, infection rates (IR), and infection fatality rates (IFR).**

| Region | Population (in millions) | COVID-19 (reported deaths) | Completeness (available data) | Total deaths (predicted) | TPR (max) | IFR in % (lower limit) | IR from DP mean (95 % cl) |
|---|---|---|---|---|---|---|---|
| Emilia-Romagna | 4.41 | 4463 | 0.97 | 6642 ± 393 | 0.29 | 0.51 | 0.15 (0.07–0.27) |
| Liguria | 1.60 | 1575 | 0.97 | 2740 ± 242 | 0.35 | 0.49 | 0.13 (0.06–0.23) |
| Lombardia | 9.86 | 16880 | 0.99 | 28989 ± 706 | 0.40 | 0.73 | 0.29 (0.15–0.52) |
| Marche | 1.57 | 987 | 0.93 | 1535 ± 207 | 0.43 | 0.23 | 0.09 (0.05–0.16) |
| Piemonte | 4.43 | 4150 | 0.96 | 6830 ± 444 | 0.36 | 0.43 | 0.13 (0.06–0.23) |
| Toscana | 3.73 | 1143 | 0.97 | 1777 ± 377 | 0.18 | 0.27 | 0.04 (0.02–0.07) |
| Veneto | 4.93 | 2128 | 0.93 | 3261 ± 376 | 0.09 | 0.73 | 0.06 (0.03–0.11) |
| Bergamo | 1.09 | 3251 | 0.99 | 6254 ± 48 | 1.00 | 0.57 | 0.72 (0.36–1.29) |

We estimate total deaths (as of 11 April 2020), lower limit IFR (by assuming IR = TPR (test positive rate) for all regions except Bergamo, for which we take IFR lower bound = PFR) and IR by normalizing to Diamond Princess (DP) IFR for age group above 70 years. Completeness is the fraction of regional population for which we have mortality data in our main dataset. The total death errors are 1 sigma errors (68% confidence interval), and 95% confidence interval for IR from DP.

Poisson distribution to estimate the errors. The population distribution in this age group on the DP was 80% in 70–79 and 20% above 80[6]. For each region of Italy, we re-weigh the population to match this age distribution and hence match the age-weighted IFR to the DP in the 70–89 age group. Combining this with the corresponding PFR, we are able to estimate IRs for this age group. Then, under the assumption of age-independent IR, we can combine this estimated IR with observed PFR for other age groups to derive IFR for all the other age groups (Table 1). IR range from 4% up to 29% (15–52% 95% CI) in Lombardia and 72% (36–100% 95% CI) in the province of Bergamo. In all cases, the estimated mean IR is below the upper limit set by the maximum TPR.

*Age dependence of IFR.* The right panel of Fig. 5 shows our estimate of these DP-anchored IFR estimates. As we make the assumption of constant IR for all age groups, we focus on the regions with high IR (>10%) where these assumptions are more likely to hold. The most reliable data come from Lombardia and Bergamo, as they are nearly complete, past the peak, and have a high number of statistics with small errors. The age-dependent IFR range from below 0.04% for ages below 50 years to 2.5%,

7.24%, and 20% for ages 70–79, 80–89, and above 90 years, respectively, (Table 2). This is broadly consistent with the estimates from the Hubei province in China, but suggests a steeper age dependence as shown in Fig. 5 for Verity et al[3,5–8]. analysis. Although the overall amplitude of our IFR estimates is anchored to DP, the relative age dependence is not.

*Crude mortality rate per year traces IFR.* In Table 2, we list the crude mortality rate per year (YMR), i.e., the fraction of the population that on average dies within a year for each age group and region. We make an interesting observation that the YMR traces the IFR for ages above 60 within 20% for different regions in Italy. One possible explanation for this trend could be that the YMR takes into account varying prevalences of co-morbidities across different population and ages. Many of the co-morbidities that reduce the general life expectancy might also lead to an increased risk of dying from COVID-19. We find that this observation holds in other places as well, for instance New York City where the population YMR is 0.62% and United Kingdom[9]. This number is similar to the NYC IFR estimated from combining the age-dependent IFR from Italy data, and consistent with a lower bound of 0.49% IFR estimated independently of Italian

**Table 2 Age distribution of fatalities and infection fatality ratios (IFR).**

| Region | Age group | Population fraction | Yearly mortality rate in % | Fraction of COVID-19 (reported deaths) | Fraction of estimated total deaths | IFR in % (lower limit) | IFR in % from DP mean (95% CI) |
|---|---|---|---|---|---|---|---|
| Lombardia population IFR (in % from DP mean (95% CI)) 0.99 (0.50–1.78) | 30–39 | 0.120 | 0.00 | 0.00 | 0.001 | 0.01 | 0.01 (0.00–0.03) |
| | 40–49 | 0.161 | 0.11 | 0.01 | 0.007 | 0.04 | 0.05 (0.02–0.09) |
| | 50–59 | 0.159 | 0.28 | 0.04 | 0.027 | 0.13 | 0.18 (0.09–0.33) |
| | 60–69 | 0.120 | 0.75 | 0.11 | 0.081 | 0.52 | 0.71 (0.35–1.28) |
| | 70–79 | 0.101 | 2.10 | 0.28 | 0.238 | 1.82 | 2.50 (1.25–4.48) |
| | 80–89 | 0.060 | 6.60 | 0.41 | 0.408 | 5.27 | 7.24 (3.60–12.96) |
| | ≥90 | 0.012 | 18.80 | 0.15 | 0.236 | 14.55 | 20.00 (9.90–35.82) |
| Bergamo population IFR (in % from DP mean (95% CI)) 0.85 (0.42–1.49) | 30–39 | 0.120 | 0.00 | 0.00 | 0.001 | 0.01 | 0.01 (0.00–0.02) |
| | 40–49 | 0.161 | 0.11 | 0.01 | 0.009 | 0.04 | 0.05 (0.02–0.09) |
| | 50–59 | 0.161 | 0.26 | 0.04 | 0.031 | 0.12 | 0.17 (0.08–0.30) |
| | 60–69 | 0.121 | 0.76 | 0.15 | 0.099 | 0.52 | 0.72 (0.35–1.29) |
| | 70–79 | 0.094 | 2.10 | 0.38 | 0.281 | 1.89 | 2.62 (1.30–4.69) |
| | 80–89 | 0.052 | 6.60 | 0.36 | 0.397 | 4.84 | 6.70 (3.33–11.99) |
| | ≥90 | 0.010 | 19.30 | 0.06 | 0.180 | 11.06 | 15.29 (7.56–27.39) |
| Emilia-Romagna population IFR (in % from DP mean (95% CI)) 0.96 (0.49–1.77) | 30–39 | 0.116 | 0.00 | 0.00 | 0.004 | 0.02 | 0.04 (0.01–0.08) |
| | 40–49 | 0.161 | 0.11 | 0.01 | 0.007 | 0.03 | 0.05 (0.01–0.10) |
| | 50–59 | 0.157 | 0.29 | 0.04 | 0.031 | 0.11 | 0.22 (0.10–0.39) |
| | 60–69 | 0.121 | 0.75 | 0.11 | 0.077 | 0.36 | 0.70 (0.34–1.26) |
| | 70–79 | 0.103 | 1.99 | 0.28 | 0.249 | 1.37 | 2.67 (1.32–4.79) |
| | 80–89 | 0.066 | 6.60 | 0.41 | 0.383 | 3.30 | 6.44 (3.17–11.54) |
| | ≥90 | 0.016 | 19.10 | 0.15 | 0.247 | 8.80 | 17.17 (8.40–30.82) |

We show the age distribution of reported COVID-19 and our estimation of excess mortality for Lombardia, Bergamo, and Emilia-Romagna, and the corresponding IFR estimates—the lower limit and estimated IFR from normalizing 70–89 IFR to Diamond Princess (DP) data, as explained in the text. The errors are small for fraction of total deaths and IFR lower limit, and we report 95% confidence interval (CI) for IFR from DP. We also show age fraction and yearly mortality for 2017: the latter traces IFR above age of 60 within 20%. The age-averaged yearly mortality rate is 0.98% for Lombardia, 0.91% for Bergamo, and 1.13% for Emilia-Romagna. We also show crude mortality rate per year, which traces IFR above age of 60 to within 20%.

data—by combining the NYC COVID-19 PFR of 0.286% as of October 2020, with the maximum TPR of 0.58% which was reached in April 2020. Similarly, as of April 22 2020, with 9900 confirmed deaths, this IFR of 0.62% predicted 19.3% IR. This is in good agreement with the then estimated IR of 23.2% for the NYC population and 16% IR for 65+ years age group from seropositivity tests[10]. Another interesting observation is that the age-dependent YMR also matches the proportion of deaths in different age groups. The proportion of YMR predicts COVID-19 mortality rate in age groups 45–64, 65–74, and above 75 to be 19%, 18%, and 55%, respectively[11]. This is in agreement with the current official NYC COVID-19 death fractions of 22%, 24%, and 50%[11]. These numbers match the higher death fraction among the younger population that is observed in the US compared with some Italian regions like Lombardia where only 8% of fatalities are in the age group <65 years of age.

## Discussion

To our knowledge, our analysis makes use of the most recent available mortality data set with the highest completeness and the least statistical uncertainties to estimate fatality rates for COVID-19. Our results suggest that a significant population of older people has died of COVID-19 without entering the official statistics. This leads to an underestimation of total deaths in Italy by more than a factor of 1.5. We note that similar excess deaths have been reported for the United States[12,13] and other countries in the Europe[14].

For policy decisions, one of the key parameters is the IFR and in this article we derived a strong lower bound from combining the PFR with the TPR. This bound is 0.73% from Lombardia and 0.50% from NYC. Our bound of 0.57% from Bergamo is independent of the IR and TPR, as we make the most conservative assumption of IR~1 to derive it.

Our work has implications on the age distribution of the mortality, which is skewed even further to the older population

than the official COVID-19 statistics suggest (see Table 2). Owing to the high number statistics, we can estimate the IFR in lower age groups very precisely: for example, we obtain 0.04% IFR (0.02–0.09% 95 % CI) in age group 40–49, a lot lower than previous estimates[5]. We also make an observation that the yearly mortality rate traces IFR in different regions of Italy and New York City, even matching the higher fraction of total fatalities attributed to the younger population (<65 years) in New York City as compared with Lombardia. This suggests that COVID-19 mortality tracks the YMR.

For this work to be globally relevant, we can test the hypothesis that the age-dependent IFR is location-independent by comparing our lower bounds and mean estimates to IFR estimates from other regions. We find that our IFR estimates are lower than the CFR estimates of most countries[15], which is as it should be since CFR is commonly taken to be an upper bound on the IFR. We note that if only symptomatic cases are being tested and assuming a 50% asymptomatic ratio as suggested by the DP data, IFR<0.5 CFR, however, for countries with high test rates (e.g., Iceland) IFR<CFR may be more applicable. For countries like Iceland with significantly different age distribution as compared with Italy, it is important to take these age-distributions into account to explain the observed differences. For instance, if we assume the IFR to track YMR, which inherently takes regional age distribution into account, we estimate the IFR lower bound (give by ~0.8× YMR) for Iceland ~0.52% (given its YMR of 0.65%[16]), as compared with the current Iceland CFR of 0.46% (0.22–0.85%, 95% CI)[15].

Another estimate of IFR to validate our lower bounds comes from serology tests, which estimate the IR and can be combined with the PFR. At the moment these tests suffer from the specificity error (false positive rate), which if not corrected can overestimate the overall IR. The largest serology survey to date has been performed in the Czech Republic, where the IR measured on a sample of 26,000 people was found to be only 0.4%. This could be an upper limit due to the poorly known false positive rate of

the test. With the current PFR of $2.5 \times 10^{-5}$ (which could be an underestimate to the true PFR if some deaths have been missed) this translates into an IFR lower bound of 0.64%, very similar to the other lower bound IFR estimates in this paper (Bergamo, Lombardia, NYC). An example of a serology study where specificity is a small correction is the study[17] in Gangelt, which finds an IR of 14%, and converts it to a 0.36–0.41% mean IFR based on 7–8 deaths[17]. However in this study, the error due to the small number of deaths has not been taken into account and assuming a Poisson distribution with 8 deaths (thus Poisson rate $\lambda = 8$) results in a 95% CI of 0.18–0.81%[18] for the IFR, which is consistent with our lower bound. The study could further be biased low by a possible underestimation of the total death count.

It is important to note that IRs can vary a lot within a single country and assumptions of an age-independent IR might not be valid in a population. An example of the latter comes from Singapore where the CFR is 0.05%[15]. This seemingly violates our lower bounds on the IFR. However, this contradiction likely arises from an in-homogeneous IR. In fact, anecdotal reports suggest that most of the infections are among the younger immigrant worker population, and the IR among the older population could be lower than the TPR. Thus, age-dependent IR could also be a limitation to our analysis. We note, however, that such age dependence is more likely for low IRs and our analysis has focused on regions with presumably high IRs of northern Italy. Furthermore, our age-dependent PFR from the province of Bergamo provides a lower limit to the IFR (Fig. 5), which is independent of the IR.

Our analysis also sheds light on the puzzle of high CFR in regions of Italy, for example, 10% in Lombardia. This high CFR can be explained by the high IR. In Lombardia, the total number of administered tests as of 8 September was 1,736,911, which is ≈17% of the population. With these tests, 5.8% of the total population was tested positive. A comparison to our estimated 23% IR suggests that the IR is five times higher than the number of positive tests. This disparity was much higher in the past at the peak of the pandemic—at the end of April 2020, IR was ~35 times higher than the reported number of positive cases. If tested cases are the most severe cases that likely required hospitalization, their fatality rate will be significantly higher than that of the overall infected population. Finally, we re-iterate the few assumptions that our analysis relies on and that we have highlighted throughout the text. To use more information in the available data over simple historical mean estimates, we employ two different statistical methods that make very different assumptions about the data—CGP makes the assumption of the data distribution being correlated Gaussian while SCM makes an assumption about the causal structure. We validate these assumptions with a cross-validation/placebo analysis, as shown in Supplementary Fig. 1. Our primary assumption in interpreting the results is that we attribute all the excess deaths to COVID-19 fatalities. To estimate the lower limits on IFR we have assumed the maximum TPR from the early period of the pandemic as a proxy of IFR and to estimate DP-anchored IFR, we have assumed age-independence of the IR in heavily infected regions like Lombardia and have also only accounted for the difference in age distribution of DP and Italy populations. The most direct way to verify our assumption regarding excess deaths is to perform COVID-19 tests on every fatality, which is currently not done in any location. Alternative explanations for excess fatalities could partly be ruled out by repeating our analysis in other regions of the world. This approach is becoming increasingly feasible as data aaare made available for some locations (NYC[13], France, Spain[14]) and preliminary analyses suggest a similar underestimation of COVID-19 deaths in other parts of the world[9,13]. Some statistics such as those published by NYC are now including probable deaths in the official counts.

## Methods

We use the total Italian mortality data from the Italian Institute of Statistics (Istat). We note that this is a data set of the total number of deaths from any cause, and that the cause of death is not available. The dates correspond to the actual day of death and not the day that it was reported. The data set is publicly available and contains the total number of daily deaths for 7357 towns in Italy for the period of 1 January to 30 June for years 2015–2020. The data set comprises the daily mortality for 21 age groups: 20 groups between the aged of 1–100 and one group for ages above 100. To reduce the statistical noise we combine the daily data into week-long periods and 10 age groups.

We then combine the data from the different towns in the same region for our analysis. The available data covers over 95% of the Italian population. The high completeness ensures that the sample has good representativeness and selection bias is limited. We assume the missing population is random and scale up the estimated mortality from this data set in proportion to the ratio of the sum-total population of the towns in our dataset with the total regional population, as per the 2010 census. To estimate the fatality rates, we will primarily rely on the most complete region of Lombardia (99% complete) and the province of Bergamo (99.5% complete).

We compare our numbers with the officially reported deaths from COVID-19[19]. This database is different from the total mortality described above and is maintained by the Italian "Protezione Civile" department. It contains the number of deaths associated with positive COVID-19 tests, and hence may be a lower bound to the true number of COVID-attributable deaths given the initial shortage of testing, especially outside of hospital setting. We assume that the age distribution of COVID-19 mortality in every region is the same as the national distribution, except for the province of Bergamo, which provides age-distribution data.

We perform a counterfactual analysis in which we compare the observed 2020 data to a prediction for mortality in 2020 in the absence of any pandemic (counterfactual), derived from models constructed from historical mortality data. In this approach, the past years are referred to as the control units and 2020 is the treated unit. A commonly used counterfactual model is the mean of the historical data. This approach falls short in two aspects. (1) It does not take into account the information that is available about the mortality in 2020 before the pandemic. If we make the conservative ansatz that Italy had no significant number of infections before February 16 (patient one was diagnosed with COVID-19 on Feb 20), this provides us with 6 weeks of mortality data in 2020 that contains valuable information about seasonal effects such as the severity of the flu season in 2020. (2) The second shortcoming of the historical mean model is that it does not capture correlations in the mortality data between weeks. We base our analysis on two different models, which account for these additional sources of information. Performing the analysis with two models that each make different assumptions about the data distribution offers a way to test how much the results are dependent on these assumptions.

Model 1 is a conditional mean with a Gaussian process (we will refer to it as CGP model). This model assumes that the data follow a Gaussian distribution and estimates its mean and covariance from historical data. The Gaussian distribution is the maximum entropy model for a probability distribution with finite mean and covariance. Owing to the limited amount of historical data available (only 5 years as control units), we cannot estimate any higher order correlations of the data distribution. A Gaussian is therefore the distribution, which best represents our knowledge of the data (according to the maximum entropy principle). Given that we are modeling a 25-week period, the data are not sufficient to estimate the full covariance, a matrix of size $25 \times 25$. We therefore impose a regularization with a principle component analysis. We find that two components explain 90% of the variance in the data, suggesting that the regularized covariance constitutes a good model for the true covariance. This full covariance captures correlation between different weeks. To make the model dependent on the early 2020 mortality rate, we condition the Gaussian on the data from the first 6 weeks. The conditional distribution is again Gaussian. We take its mean to be the counterfactual prediction and its variance as an error estimate.

Model 2 is a SCM. This model makes minimal assumptions about the underlying data distribution, instead, it makes assumptions about the causal structure of the data: that there is a weighted average of the control units (i.e., previous years) which predicts the potential outcome of the treated unit (current year) in the absence of the pandemic. The weights for the different control units in this linear combination are estimated by minimizing the difference between this prediction and the observed data for 2020 in the pre-pandemic period. Another implicit assumption made in SCM is that the similarities between the years, which lead to a good fit in weeks before the pandemic continue to exist in the later weeks, meaning that the same weights can be used for predicting the counterfactual in the pandemic period. It is a conservative assumption since there is no a-priori reason for why the mortality rate in the later months of 2020 should have been different from previous years (and the variations between previous years) without the pandemic. The SCM method for counterfactual analysis is well established in the social sciences and policy making[20,21].

We fit both methods on the data for each province, region, and age group. The technical details of both methods can be found in the supplementary materials and results for different age groups are presented in Supplementary Fig. 2–5. To validate our methodology, we also do a cross-validation/placebo analysis[20] in which we treat each of the control units as a treated unit in turn and use the remaining control units to predict the outcome. As in this case, the assumed treated year has not undergone any pandemic, our prediction should match the observed data and we do indeed find that overall, as shown in Supplementary Fig. 1, predictions from our proposed methods are closer to the observed data in comparison with a simple historical mean.

**Reporting summary**. Further information on research design is available in the Nature Research Reporting Summary linked to this article.

## Data availability
The processed data used in this analysis is available on GitHub[22] at https://github.com/bccp/covid-19-data and the original data can be obtained from the Italian Statistical institute https://www.istat.it/it/files/2020/03/Dataset-decessi-comunali-giornalieri-e-tracciato-record_al30giugno.zip.

## Code availability
The code is available at Github[22] and at Zenodo[23].

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

## Acknowledgements

We thank Ehud Altman, Alex Krolewski, Zarija Lukić, Jasjeet Sekhon, and Jacob Steinhardt for helpful comments and the Italian Institute of Statistics (Istat), and in particular Antonella Ciccarese, for their prompt responses and for making the data available on a short timescale.

## Author contributions

U.S., S.F., C.M., and V.B. designed the research and interpreted results. C.M. and V.B. did the main data analysis in consultation with U.S and S.F. S.F., C.M., G.S., and U.S. gathered data sets that C.M. and G.S. cleaned, and G.S. validated. All authors wrote and reviewed the manuscript.

## Competing interests
The authors declare no competing interests.
