## [Peer Review File · Nature Communications]

REVIEWER COMMENTS

Reviewer #1 (Remarks to the Author):

This paper tackles an interesting and very topical question. I found the descriptions a little vague or confusing at times. In particular, more details regarding the data sources used would be hugely useful. The results are probably fairly robust to the assumptions made, but I feel the presentation makes a number of fairly stringent statistical assumptions without really exploring their impact. The authors' interpretation seems a little stronger than justified by the data at times.

Major comments

1. Jargon. In general, I find that this manuscript is full of field-specific jargon. Nature Communications is aimed at a multidisciplinary audience. I think that this manuscript could be written in a way that is much more comprehensible to people outside the field of data science.

2. Description of statistical methods. Many of the statistical methods and assumptions are described in a confusing or vague way.

For example, Y is described as "mortality". It would help to know what aspect of mortality exactly is contained in Y . From the subsequent paper, I assume that Y contains the absolute number of all-cause deaths, but this is not stated when Y is introduced as notation.

Splitting Y into X and Z is a little confusing at first glance since X and Z are typically used to denote covariates that relate to outcomes. This is not a problem as long as the notation is clearly defined.

It appears that the authors assume that $X_{1P} = X_1$, i.e. the observed mortality in 2020 pre-February 16, is equal to the mortality that would have been experienced in the absence of the pandemic. It would be helpful if the authors explicitly stated this assumption.

Equation 1 contains the term " t " which is undefined.

The description of the Gaussian model (CGP) is confusing. Equation 2 appears to be a standard linear regression of Z_0 on X_0 , used to predict Z_1 from X_{1P} . That being so, I cannot understand how the principal components analysis fits in. From what is written, I have no real sense of how these predictions were obtained. Perhaps clearer notation would help clarify what fed into which models.

The description of the synthetic control method is similarly confusing. The method is in fact incredibly simple. The six years are used to provide 6 time-series over the 14 weeks. Pre-Feb-16, all 6 time-series are "untreated" and post-Feb-16, one of the time-series (2020) is "treated". You look for the set of weights (5 weights summing to 1) which, when used to obtain the weighted average of mortality counts during pre-16 periods among the 5 pre-2020 time-series, most closely mirror the pattern of mortality counts of the 2020 year. I think that the method could be much more clearly explained, perhaps with reference to Figure 1.

For the synthetic control method you have to assume that (i) there exists a weighted combination that adequately captures the mortality pattern pre-Feb-16, and (ii) these similarities between years continue post-Feb-16. (i) is a little uncertain here, since 2020 appears to be a slight outlier, in having particularly low mortality early in the year. (ii) is an unverifiable assumption.

3. Data used. It would be helpful for the reader to know something about the mortality data feeding into these statistics. Most importantly, it is not clear from what is written whether the COVID-19 death data and the all-cause mortality data come from the same source or not.

Secondly, for the COVID-19 death data, how is the data collected? How is a "COVID-19 death"

defined in these data? Deaths with a positive test? Deaths with positive test or clinically-diagnosed/suspected COVID? Is there a lag on data and, if so, are deaths included on the day they were reported or recorded? This sort of detail would be hugely useful in interpreting results.

For the all-cause mortality data, which towns are included in these data and why? How are towns selected for inclusion? Is there likely to be any bias arising from selection into the data?

Analyses all appear to be based on absolute numbers of deaths. This being so, it seems relevant to know about the base population in terms of absolute numbers and age/sex distribution. Are these relatively stable over the 6 years considered? If not, perhaps the authors should consider an analysis that takes these aspects into consideration (e.g. analyze age-and-sex-standardised rates).

"The scaling for this dataset is determined by matching the estimated excess mortality with that of more complete Class-2 dataset for the period of February 22-March 28." This could be explained more. Without more information it is very unclear to me what was done.

3. Interpretation of results. The authors argue strongly that any excess deaths must be due to COVID-19. They dismiss the possibility that mortality may be different in 2020 for reasons associated with the pandemic other than direct death from COVID-19. However, it has been argued elsewhere that excess deaths caused by cancelled or postponed healthcare procedures are likely to be substantial. The authors' argument, that if this were true then it would only affect regions with a lot of infection, seems a bit simplistic. Maybe more detail about Italy's response to the pandemic would clarify this point. Without knowing anything about Italy in particular, I would have thought that precautions were taken in the whole country, including patients being less likely to seek healthcare, from fairly early in the pandemic.

Here, it is also relevant to think about differences in the two reporting systems. If the all-cause mortality and COVID-19 death data come from different sources, then are there other systematic differences that might affect the comparison? E.g. reporting delays. The COVID-19 death data appears to be more complete than the all-cause mortality data, requiring some sort of scaling-up of the all-cause estimates. How robust are the comparisons to the assumptions made in that scaling-up process?

4. Estimated IFR. In order to understand the use of the Test Positivity Rate (TPR), the fraction of positive to total tests, as an estimate of the fraction of infected, more detail is needed about who (& why?) was tested in Italy during this period. Was testing largely restricted to symptomatic patients? If so, the TPR will be a huge overestimate of the infection rate. This is borne out by the fact that in Table 1 the TPR is higher than the upper CI for most of the IRs (infection rates) inferred from DP.

That being so, how useful is the IFR (infection fatality rate) estimated based on an underestimated IR? The authors do say it is a lower bound. But it may be a very low lower bound.

The alternative IFR (Table 2) is estimated using external data (DP) combined with the internal PFR estimates. This relies on an assumption of the infection rate being the same across all age groups. Is this a reasonable assumption? The authors say that "While the overall amplitude of our IFR estimates is anchored to DP, the relative age dependence of IFR is not." This seems to me to be true only if the constant-IR age assumption holds. Further, this IFR analysis assumes that the IFR on the DP is the same as the IFR in Italy, within age-groups. Again, this is a simplifying assumption. It requires, for example, the distribution of comorbidities to be similar in the two groups.

5. YMR model. The authors propose a model relating infection fatality rate to yearly mortality rate, $YMR = K IFR$, with K constant. The appropriateness of this model relies on very scant data. It is an interesting hypothesis, but it is far from being demonstrated that this relationship holds in any other setting.

6. Statistical uncertainty. It is not always clear whether statistical uncertainty has been propagated through all the models and estimates appropriately. The uncertainty is not always shown in tables and figures appropriately. The IFR in tables 1 and 2, for example, needs confidence limits.

Important to estimate IFR. However, based on a model for PFR, which is much higher than official states. What is the IFR is use reported deaths, for comparison

The overall IFR is perhaps not that useful – likely to vary massively by age and comorbidity.

Fig 1 – it would be more standard to display a 2 sigma interval (approx. corresponding to 95% confidence limits) rather than the 1 sigma interval shown.

Fig 2 – shows a long period of extrapolation beyond the data. It would probably be wise to highlight this a little more strongly.

Fig 3 – the method of predicting excess deaths does not, as far as I can tell, incorporate information on the age distribution. Therefore, what assumptions are required in order to be able to validly split by age in this way?

Minor comments

When Figure 1 is first introduced, it would be good to mention why there is huge variability across the 6 years in mortality in Jan but not Feb-April (presumably the severity of the flu season).

In Figure 1, the green line is said to fit the observed data by construction. But it doesn't fit the observed data. Could the authors clarify this?

When Figure 2 is introduced it would be helpful to mention that the graph now just shows a smaller number of regions, and why.

Several of the graphs have confusing legends, where the different lines are explained in different plots. It would be better to have a single legend per figure containing the whole legend.

I would suggest the authors do not use pink to represent the past historical data in Fig 1 but current observed data in Fig 2.

The "conservative assumption" in Figure 2(a) – the authors say they assume that "the weekly excess mortality is the same as the reported COVID-19 deaths". From the graph, it appears that they mean that they assume there is no excess mortality (i.e. the total mortality is equal to the reported deaths). This is very unclear as written.

Reviewer #2 (Remarks to the Author):

This is an interesting COVID-19 study and I expect many will be interested in mortality patterns over time, by age, and geography in Italian towns.

Overall, there has been additional work on excess mortality since (<https://pubmed.ncbi.nlm.nih.gov/32407306/>, <https://www.medrxiv.org/content/10.1101/2020.04.15.20066431v2>, <https://www.euromomo.eu/graphs-and-maps/>) which would be useful to integrate more fully in

the analysis and discussion.

Specific comments:

1) The excess mortality approach is interesting, although it is not the classical time series approach used in epidemiology. It would be useful to compare your excess mortality estimates with those of time series approaches using data for the entire year (eg, NYC would make for a good comparison <https://pubmed.ncbi.nlm.nih.gov/32407306/>, or the EuroMomo estimates <https://www.euromomo.eu/graphs-and-maps/>)

2) There is some discussion of completeness in the data section, and upscaling observations for lack of completeness. It is unclear how completeness is established here – is this based on past years, and if so could completeness be different in 2020 because of the pandemic? We have seen this happen in the US. Are these meant to capture reporting delays, or deaths that will truly never be captured by the system?

3) Can the authors generate excess mortality patterns in the under 40 yrs old?

4) Fig 1 is reassuring in that the baseline fits regional observations in the pre-pandemic period. It would be useful to provide a similar plot for age-specific mortality patterns in main text or supplement.

5) Comments/suggestions about interpretation of results:

a. Authors explore lags between excess mortality and official COVID-19 tallies and find a 6-day lag. They interpret this as possibly a consequence of hospital treatment postponing death by several days on average. I don't get this. Are both datasets by date of death, or date of report? Could it be a longer reporting delay in hospital deaths? Doesn't the interpretation also depend on the proportion of deaths captured by hospital surveillance, vs deaths occurring untested at home or in nursing homes? This section needs to be elaborated on.

b. Regional differences and "true" attribution of deaths due to COVID-19 (Fig 2 and from text: "COVID-19 has put an enormous pressure on Italy's medical system and social services. This could have led to fatalities that could otherwise be averted, causing us to overestimate the COVID-19 deaths. However, the pressure on the medical system is regional and likely sustainable for regions with a low number of official COVID-19 deaths. Instead, we consistently find a very large excess in mortality in most of the regions in Italy, including those that reported nearly zero COVID-19 deaths." I disagree; many regions (from Fig 1) did not see excess deaths. This is a very interesting question, but difficult to conclude with the info provided here. It would also be useful to see excess death rates per population, rather than numbers, to gauge the intensity of the epidemic in different regions. Overall, a less handwavy reasoning would be useful here (ie, is there a stable ratio of cumulative excess deaths to official deaths between regions, and hence no indication that a collapse in the health care system is to blame for mortality rather than the virus?)

6) Consider updating results with most recent mortality data

Response to Referees

We thank the referee for useful comments. In this document, we have addressed each point raised by the referee- giving further clarifications, rebuttal and pointing out the changes we have made in the manuscript, as appropriate.

The referee comments are in black and our responses are in red.
The changes we have made in the main manuscript are in blue.

In addition to making changes to address referee comments, we have also updated our analysis with most recent and complete data. This makes our submission much stronger and we hope to have addressed all the comments to the referee's satisfaction for acceptance.

Thank you

Reviewer #1 (Remarks to the Author):

This paper tackles an interesting and very topical question. I found the descriptions a little vague or confusing at times. In particular, more details regarding the data sources used would be hugely useful. The results are probably fairly robust to the assumptions made, but I feel the presentation makes a number of fairly stringent statistical assumptions without really exploring their impact. The authors' interpretation seems a little stronger than justified by the data at times.

Major comments

1. Jargon. In general, I find that this manuscript is full of field-specific jargon. Nature Communications is aimed at a multidisciplinary audience. I think that this manuscript could be written in a way that is much more comprehensible to people outside the field of data science.

We have updated the manuscript explicitly defining the specific terms used. We believe it is now better accessible to a broad audience.

2. Description of statistical methods. Many of the statistical methods and assumptions are described in a confusing or vague way.

For example, Y is described as "mortality". It would help to know what aspect of mortality exactly is contained in Y. From the subsequent paper, I assume that Y contains the absolute number of all-cause deaths, but this is not stated when Y is introduced as notation.

We have clarified the meaning of "Y", as the total number of deaths from any cause.

Splitting Y into X and Z is a little confusing at first glance since X and Z are typically used to denote covariates that relate to outcomes. This is not a problem as long as the notation is clearly defined.

It appears that the authors assume that $X1P = X1$, i.e. the observed mortality in 2020 pre-February 16, is equal to the mortality that would have been experienced in the absence of the pandemic. It would be helpful if the authors explicitly stated this assumption.

We have made the assumption clear in the text.

Equation 1 contains the term “t” which is undefined.

We have removed the subscript “t” since it’s unnecessary here and have simplified the notation.

The description of the Gaussian model (CGP) is confusing. Equation 2 appears to be a standard linear regression of $Z0$ on $X0$, used to predict $Z1$ from $X1P$. That being so, I cannot understand how the principal components analysis fits in. From what is written, I have no real sense of how these predictions were obtained. Perhaps clearer notation would help clarify what fed into which models.

We added more explanatory text.

The description of the synthetic control method is similarly confusing. The method is in fact incredibly simple. The six years are used to provide 6 time-series over the 14 weeks. Pre-Feb-16, all 6 time-series are “untreated” and post-Feb-16, one of the time-series (2020) is “treated”. You look for the set of weights (5 weights summing to 1) which, when used to obtain the weighted average of mortality counts during pre-16 periods among the 5 pre-2020 time-series, most closely mirror the pattern of mortality counts of the 2020 year. I think that the method could be much more clearly explained, perhaps with reference to Figure 1.

We thank the referee for the suggestion, and we have indeed added an explanation of SCM in these terms.

For the synthetic control method you have to assume that (i) there exists a weighted combination that adequately captures the mortality pattern pre-Feb-16, and (ii) these similarities between years continue post-Feb-16. (i) is a little uncertain here, since 2020 appears to be a slight outlier, in having particularly low mortality early in the year. (ii) is an unverifiable assumption.

These are certainly assumptions that we need to make. Regarding (i), the SCM results pre-pandemic are rather close to the 2020 data, meaning that there is a linear combination of previous years that match the pre-pandemic part of 2020. For (ii), we agree that it’s an assumption of SCM, and that’s part of the reason why we also try a different CGP method with different assumptions and show that they give consistent results.

3. Data used. It would be helpful for the reader to know something about the mortality data feeding into these statistics. Most importantly, it is not clear from what is written whether the COVID-19 death data and the all-cause mortality data come from the same source or not.

We have clarified that they come from different government agencies in Italy, and we expanded the description of the datasets.

Secondly, for the COVID-19 death data, how is the data collected? How is a “COVID-19 death” defined in these data? Deaths with a positive test? Deaths with positive test or clinically-diagnosed/suspected COVID? Is there a lag on data and, if so, are deaths included on the day they were reported or recorded? This sort of detail would be hugely useful in interpreting results.

It is deaths with a positive COVID-19 test -- we don't know the exact details of who got tested especially in the early days of the pandemic due to the severe lack of tests, but this is evident in the smaller reported number of official COVID-19 deaths compared to the excess mortality (our estimate of the actual true COVID-19 deaths). Note that we only use the “official” COVID-19 deaths to show that they were underreported initially, while our other conclusions are independent of this data.

The date refers to the day of death, so there is no “delay”. Further details can be found on the official website <https://github.com/pcm-dpc/COVID-19> and we mention this in the text.

For the all-cause mortality data, which towns are included in these data and why? How are towns selected for inclusion? Is there likely to be any bias arising from selection into the data?

The current analysis is based on 7357 towns out of 7904 total towns, covering more than 95% of the Italian populations (corresponding to the towns that reported the mortality to ISTAT, the national institute of statistics), and over 99% complete for Lombardia which is one of the main focuses of the paper. Any biases, if present at all, should be negligible given the high completeness.

Analyses all appear to be based on absolute numbers of deaths. This being so, it seems relevant to know about the base population in terms of absolute numbers and age/sex distribution. Are these relatively stable over the 6 years considered? If not, perhaps the authors should consider an analysis that takes these aspects into consideration (e.g. analyze age-and-sex-standardised rates).

Between 2016 and 2020 the Italian population changed by 0.3%, and the relative demographics changed by less than 0.5%.

“The scaling for this dataset is determined by matching the estimated excess mortality with that of more complete Class-2 dataset for the period of February 22-March 28.” This could be explained more. Without more information it is very unclear to me what was done.

With the new dataset that was released after the initial submission, we don't need to perform this procedure anymore, because we only have one class of dataset for the whole period.

3. Interpretation of results. The authors argue strongly that any excess deaths must be due to COVID-19. They dismiss the possibility that mortality may be different in 2020 for reasons associated with the pandemic other than direct death from COVID-19. However, it has been argued elsewhere that excess deaths caused by cancelled or postponed healthcare procedures are likely to be substantial. The authors' argument, that if this were true then it would only affect regions with a lot of infection, seems a bit simplistic. Maybe more detail about Italy's response to the pandemic would clarify this point. Without knowing anything about Italy in particular, I would have thought that precautions were taken in the whole country, including patients being less likely to seek healthcare, from fairly early in the pandemic.

There has been a lot of speculation on this issue. Mortality could have been lower because of the lockdowns (fewer accidents etc), or higher because of postponed healthcare procedures. We simply point out that the excess mortality we observe happened mainly at the beginning of the pandemic, far exceeding the regular mortality rates per unit time. Anecdotal evidence from the hardest hit regions suggests many of the deaths occurred in retirement homes without even getting to seek hospital care.

Here, it is also relevant to think about differences in the two reporting systems. If the all-cause mortality and COVID-19 death data come from different sources, then are there other systematic differences that might affect the comparison? E.g. reporting delays. The COVID-19 death data appears to be more complete than the all-cause mortality data, requiring some sort of scaling-up of the all-cause estimates. How robust are the comparisons to the assumptions made in that scaling-up process?

Completeness is no longer an issue, since we now have 95% completeness, while for Lombardia it's over 99%. The conclusions are however unchanged from the previous analysis, but the new data allows us to make fewer assumptions.

4. Estimated IFR. In order to understand the use of the Test Positivity Rate (TPR), the fraction of positive to total tests, as an estimate of the fraction of infected, more detail is needed about who (& why?) was tested in Italy during this period. Was testing largely restricted to symptomatic patients? If so, the TPR will be a huge overestimate of the infection rate. This is borne out by the fact that in Table 1 the TPR is higher than the upper CI for most of the IRs (infection rates) inferred from DP.

That being so, how useful is the IFR (infection fatality rate) estimated based on an underestimated IR? The authors do say it is a lower bound. But it may be a very low lower bound.

We agree with the referee that this is a lower bound, but it is not much lower than our best estimate of the IFR, and our lower bound is higher than some other claims for IFR. We believe this lower bound is useful, specially in light of recent speculations of a high IR worldwide (India, US). Even our lower bound is in conflict with a high IR and the observed mortality, and if our lower bound was a very low bound the conflict only grows. So for example the Bergamo lower bound IFR of 0.57%, which is based on the fraction of actual deaths in overall population (hence a robust lower bound), puts a strong constraint on some of these speculations.

The alternative IFR (Table 2) is estimated using external data (DP) combined with the internal PFR estimates. This relies on an assumption of the infection rate being the same across all age groups. Is this a reasonable assumption? The authors say that “While the overall amplitude of our IFR estimates is anchored to DP, the relative age dependence of IFR is not.” This seems to me to be true only if the constant-IR age assumption holds. Further, this IFR analysis assumes that the IFR on the DP is the same as the IFR in Italy, within age-groups. Again, this is a simplifying assumption. It requires, for example, the distribution of comorbidities to be similar in the two groups.

We agree with the referee and state this constant IR assumption explicitly in the “Results” section, when discussing the age dependence of IFR. We believe this to be a reasonable assumption in the early days of the pandemic when no age group was taking precautions. Moreover, our main conclusions are based on regions with very high total infection rate, further supporting the assumption. We agree that some of these assumptions may not hold today, as there is evidence of a lowering of the age of the average Covid-positive patients, indicating a higher IR among younger people.

We also agree with the referee on other simplifying assumptions made and we state them explicitly now in the DP section. We believe the similarity of DP analysis to the lower bound analysis is telling.

5. YMR model. The authors propose a model relating infection fatality rate to yearly mortality rate, $YMR = K IFR$, with K constant. The appropriateness of this model relies on very scant data. It is an interesting hypothesis, but it is far from being demonstrated that this relationship holds in any other setting.

We agree with the referee that is based on circumstantial evidence and not necessarily a fundamental law governing the virus, however we are merely pointing out that this approximate relation holds with $K \sim 1$ for the initial stages of the pandemic when limited therapeutic intervention was possible. With modern therapeutics and support, we expect the IFR to be reduced, and therefore we put less emphasis on this observation in the revised version of the paper. However, this YMR IFR relation does explain some puzzling observations, such as the age distribution of fatalities in NYC vs Italy, and seems a better null hypothesis than no IFR adjustment for age distribution and comorbidities at all.

6. Statistical uncertainty. It is not always clear whether statistical uncertainty has been propagated through all the models and estimates appropriately. The uncertainty is not always

shown in tables and figures appropriately. The IFR in tables 1 and 2, for example, needs confidence limits.

We show this in Figure 5, but since these are lower limit, we decided to just report the number in the tables.

The overall IFR is perhaps not that useful – likely to vary massively by age and comorbidity.

We agree with the referee, but this is a number that is often quoted. We do stratify by age, but don't have the data necessary to do the same by co-morbidity. However, the relation between IFR and YMR, while likely to change in time because of improved treatments, also allows one to quickly adjust for population age distribution and comorbidity. For example, YMR in US is 0.6% vs 1% in Italy, and a similar ratio should apply to IFR. We believe this YMR based correction, while speculative, is better than no correction at all, where IFR is often quoted without reference to the underlying age distribution and comorbidity factors.

Fig 1 – it would be more standard to display a 2 sigma interval (approx. corresponding to 95% confidence limits) rather than the 1 sigma interval shown.

Fig 2 – shows a long period of extrapolation beyond the data. It would probably be wise to highlight this a little more strongly.

The period of extrapolation is considerably shorter in the new version of the paper and we highlighted it explicitly.

Fig 3 – the method of predicting excess deaths does not, as far as I can tell, incorporate information on the age distribution. Therefore, what assumptions are required in order to be able to validly split by age in this way?

To estimate the excess deaths for different age groups, we need the age distribution of the total mortality which the data provides us and the age distribution of the official COVID-19 deaths. This latter distribution is unavailable for different regions, but is available nationally. Hence we assume that this distribution remains the same across regions. We state this explicitly in the Data section.

Minor comments

When Figure 1 is first introduced, it would be good to mention why there is huge variability across the 6 years in mortality in Jan but not Feb-April (presumably the severity of the flu season).

This was due to a different time period for the first week of the year to align with COVID data and we have corrected it in the current figure.

In Figure 1, the green line is said to fit the observed data by construction. But it doesn't fit the observed data. Could the authors clarify this?

The green line coincides with the observed data by construction up to the last data point before the vertical line which marks the onset of the pandemic. At this point it becomes a prediction and is allowed to deviate.

When Figure 2 is introduced it would be helpful to mention that the graph now just shows a smaller number of regions, and why.

We focus only on the hardest regions since they lead to most robust conclusions and we have added the sentence stating it in the text when we introduce Figure 2.

Several of the graphs have confusing legends, where the different lines are explained in different plots. It would be better to have a single legend per figure containing the whole legend.

We thank the referee for the suggestion. We tried this but we were unable to have the plots look clearer than they are now, due to the number of items in the legend.

I would suggest the authors do not use pink to represent the past historical data in Fig 1 but current observed data in Fig 2.

We have updated Fig 1 to not use pink.

The “conservative assumption” in Figure 2(a) – the authors say they assume that “the weekly excess mortality is the same as the reported COVID-19 deaths”. From the graph, it appears that they mean that they assume there is no excess mortality (i.e. the total mortality is equal to the reported deaths). This is very unclear as written.

We thank the referee for pointing this out and have now clarified this statement in the caption of the figure.

Reviewer #2 (Remarks to the Author):

This is an interesting COVID-19 study and I expect many will be interested in mortality patterns over time, by age, and geography in Italian towns.

Overall, there has been additional work on excess mortality since (<https://pubmed.ncbi.nlm.nih.gov/32407306/>,

<https://www.medrxiv.org/content/10.1101/2020.04.15.20066431v2>,
<https://www.euromomo.eu/graphs-and-maps/>) which would be useful to integrate more fully in the analysis and discussion.

We have now included these in our discussion.

Specific comments:

1) The excess mortality approach is interesting, although it is not the classical time series approach used in epidemiology. It would be useful to compare your excess mortality estimates with those of time series approaches using data for the entire year (eg, NYC would make for a good comparison <https://pubmed.ncbi.nlm.nih.gov/32407306/>, or the EuroMomo estimates <https://www.euromomo.eu/graphs-and-maps/>)

The two methods that we use are completely data-driven and are state of the art in the social and econometric sciences, and we believe are very useful and applicable in this context. A general comparison between different methods would indeed be very useful, but it's beyond the scope of this paper, so here we show that two independent methods give very similar results.

2) There is some discussion of completeness in the data section, and upscaling observations for lack of completeness. It is unclear how completeness is established here – is this based on past years, and if so could completeness be different in 2020 because of the pandemic? We have seen this happen in the US. Are these meant to capture reporting delays, or deaths that will truly never be captured by the system?

With the new and improved data that we use in the revised manuscript, the population completeness is over 95% for all of Italy, and over 99% in the most hard hit regions (such as Lombardia) which drive our conclusions. Therefore no rescaling or assumptions are needed in the revised version.

3) Can the authors generate excess mortality patterns in the under 40 yrs old?

As Fig. 3 shows, the estimated excess mortality is already only marginally significant in the age group 40-49. We checked and found no significant excess in the lower age groups. Because of constraints on the length of the manuscript we decided not to show these figures.

4) Fig 1 is reassuring in that the baseline fits regional observations in the pre-pandemic period. It would be useful to provide a similar plot for age-specific mortality patterns in main text or supplement.

We have added this plot to the supplement, since we have no space in the main text.

5) Comments/suggestions about interpretation of results:

a. Authors explore lags between excess mortality and official COVID-19 tallies and find a 6-day lag. They interpret this as possibly a consequence of hospital treatment postponing death by several days on average. I don't get this. Are both datasets by date of death, or date of report? Could it be a longer reporting delay in hospital deaths? Doesn't the interpretation also depend on the proportion of deaths captured by hospital surveillance, vs deaths occurring untested at home or in nursing homes? This section needs to be elaborated on.

Both data sets are by date of death. The primary objective of the correlation analysis was to establish a correlation between the two data sets. The time lag emerged as a finding of this analysis. The delay due to treatment is only one possible explanation for this. It could also be due to a ramping up of testing with time. We have clarified this section.

b. Regional differences and "true" attribution of deaths due to COVID-19 (Fig 2 and from text: "COVID-19 has put an enormous pressure on Italy's medical system and social services. This could have led to fatalities that could otherwise be averted, causing us to overestimate the COVID-19 deaths. However, the pressure on the medical system is regional and likely sustainable for regions with a low number of official COVID-19 deaths. Instead, we consistently find a very large excess in mortality in most of the regions in Italy, including those that reported nearly zero COVID-19 deaths.")

I disagree; many regions (from Fig 1) did not see excess deaths. This is a very interesting question, but difficult to conclude with the info provided here. It would also be useful to see excess death rates per population, rather than numbers, to gauge the intensity of the epidemic in different regions. Overall, a less handwavy reasoning would be useful here (ie, is there a stable ratio of cumulative excess deaths to official deaths between regions, and hence no indication that a collapse in the health care system is to blame for mortality rather than the virus?)

We agree with the referee that we can't make strong conclusions given the data available, however we show in Figure 5 (green line) that the fraction of deaths "missed" by the official Covid counts is similar throughout the pandemic, starting from close to 1 and gradually decreasing with time, for all of the regions considered. While the error bars are somewhat large, we see no indication that additional factors beyond Covid-19 are at play here. We have rephrased the paragraph, clarifying it.

6) Consider updating results with most recent mortality data

We have updated the analysis with the most recent data, which now extends to the end of the first wave in Italy, and also have much higher completeness, so that fewer assumptions are needed.

REVIEWER COMMENTS

Reviewer #1 (Remarks to the Author):

The revised manuscript is clearer, particularly with regard to the data used.

The assumptions underlying the different models are a bit more explicit, but still downplayed. For example, SCM is said to make "minimal assumptions about the underlying data distribution". What is not said, is that the SCM makes, in parallel, strong assumptions about the causal structure. The data can follow any distribution, but you're still assuming that the weighted combination which predicts the outcome pre-pandemic would continue to do so after that in the absence of a pandemic; this is in no way a trivial assumption. Subsequently the authors say the SCM and CGP provide better counterfactuals than historical means and make "more accurate and precise predictions". This may be true, but it's not guaranteed; it depends how well the assumptions are satisfied. Again, I think nuance is lacking in the interpretation. Overall, I think the authors downplay the extent of the underlying assumptions. The concluding paragraph starts with: "Our analysis relies on a few assumptions that we have highlighted throughout the text. Primarily, we attribute all the excess deaths to COVID-19 fatalities." This seems to imply that there are no other really important assumptions required, but there really are. I would reword this.

I still have concerns about the YMR model. In my opinion this is merely an interesting observation rather than a "model". I think its validity is very overstated. For instance, the abstract proposes it as a model for estimating the IFR for countries across the world. This puts far too much emphasis on a possible chance correlation in an unvalidated model.

In response to the other reviewer, the authors have added the statement: "we consistently find a similar and very large excess in mortality over the official counts in most of the regions in Italy, which appears to be independent of how hard the region was hit." But, as pointed out by the other reviewer, Figure 1 seems to show that many regions in Italy do not appear to experience a large excess in mortality. So could the authors explain that apparent contradiction (in the text).

In respect to the jargon, the current manuscript is a little better, but still not written for what I believe the Nature Communication audience to be. Some specific remaining examples are below:

- "Estimating covariance matrix in GP" – what is GP?
- The repeated use of the word "bin", where group or category would do just as well.
- I don't think the quantities CFR, IFR and PFR are defined anywhere. (We're told CFR means case fatality rate, but it would be best to define what that quantity actually is, rather than just the words the acronym stands for).
- "1 and 2sigma confidence interval" And "1-sigma confidence interval". It is standard to give the level of the confidence interval, e.g. 95% confidence interval. This would be best, but if the sigma terminology is retained, the authors should tell the reader what sigma is.
- "cl" as shorthand for a confidence interval. Again, I couldn't find this defined anywhere.

Response to the referee

We thank the referee for the feedback. We have made major revisions to the text and have run additional tests to address the concerns that were raised. Specifically we have rewritten the methods section completely. It now focuses on the basic assumptions that enter our models and their justification. Technical details and jargon have been moved to the supplemental material. We have further performed an additional placebo analysis to test the validity of the models. The results of this analysis have been added to the supplemental material, too. The results support the suitability of our approach and are discussed further below.

Major changes in the text are highlighted in **blue**.

We would like to point out that some of the points raised by the referee were already included in the current version, specifically the assumptions that enter the SCM, the apparent contradiction of Fig. 1, and the definitions of acronyms like IFR, PFR and CFR.

However we understand the referee's concern that they might be overlooked and hence strived to emphasize them more. Our point-wise response to the referee's comments (in red) is below (in black).

The assumptions underlying the different models are a bit more explicit, but still downplayed. For example, SCM is said to make "minimal assumptions about the underlying data distribution". What is not said, is that the SCM makes, in parallel, strong assumptions about the causal structure. The data can follow any distribution, but you're still assuming that the weighted combination which predicts the outcome pre-pandemic would continue to do so after that in the absence of a pandemic; this is in no way a trivial assumption. Subsequently the authors say the SCM and CGP provide better counterfactuals than historical means and make "more accurate and precise predictions". This may be true, but it's not guaranteed; it depends how well the assumptions are satisfied. Again, I think nuance is lacking in the interpretation. Overall, I think the authors downplay the extent of the underlying assumptions. The concluding paragraph starts with: "Our analysis relies on a few assumptions that we have highlighted throughout the text. Primarily, we attribute all the excess deaths to COVID-19 fatalities." This seems to imply that there are no other really important assumptions required, but there really are. I would reword this.

We completely reworded the method section to highlight the underlying assumptions of both methods. These assumptions are then explicitly repeated in the last paragraph of the discussion section. Instead of simply referring to other sections, we now also repeat other important assumptions, such as those made to estimate the lower limit of the IFR and the Diamond Princess analysis, in the discussion section.

To address the concerns regarding the significance of the assumptions that enter the SCM and CGP methods, we have performed a cross-validation (placebo) analysis on the control units. In this analysis, we treat one treated unit at a time as a placebo treated unit and predict its outcome with models constructed from the remaining control units. We find that overall, the predictions from our methods are a close match to the observed data. The mean error between

these predictions and the observed data is also smaller for SCM or CGP than for the mean model. The errors (deviations) measured from this analysis include the assumptions of the model. We have added this analysis to the supplementary material. We also point out that the CGP also directly provides an error estimate which informs what is statistically significant.

I still have concerns about the YMR model. In my opinion this is merely an interesting observation rather than a “model”. I think its validity is very overstated. For instance, the abstract proposes it as a model for estimating the IFR for countries across the world. This puts far too much emphasis on a possible chance correlation in an unvalidated model.

We have re-written the YMR section and modified the text in the abstract and the discussion. It now simply states that this is an interesting observation that seems to hold in Italy and NYC. We do not refer to it as a model anymore.

In response to the other reviewer, the authors have added the statement: “we consistently find a similar and very large excess in mortality over the official counts in most of the regions in Italy, which appears to be independent of how hard the region was hit.” But, as pointed out by the other reviewer, Figure 1 seems to show that many regions in Italy do not appear to experience a large excess in mortality. So could the authors explain that apparent contradiction (in the text).

We now state that we observe this excess in “*many* regions in Italy”. This is shown in Figure 2 for 8 regions which were hit at different levels: Lombardia was the worst affected and Toscana or Marche not as much. We now refer to this figure in that sentence for clarification.

In respect to the jargon, the current manuscript is a little better, but still not written for what I believe the Nature Communication audience to be.

We have given the method section a complete rewrite.

Some specific remaining examples are below:

- “Estimating covariance matrix in GP” – what is GP?
We now refer to it as Gaussian Process explicitly, and have moved these technicalities to the supplementary material/
- The repeated use of the word “bin”, where group or category would do just as well.
We could find this word being used only twice and have rephrased both now.
- I don’t think the quantities CFR, IFR and PFR are defined anywhere. (We’re told CFR means case fatality rate, but it would be best to define what that quantity actually is, rather than just the words the acronym stands for).
All three of these terms were already defined in the footnotes on their first occurrence in the introduction. We have now moved the definition from footnotes into the main text.

- “1 and 2sigma confidence interval” And “1-sigma confidence interval”. It is standard to give the level of the confidence interval, e.g. 95% confidence interval. This would be best, but if the sigma terminology is retained, the authors should tell the reader what sigma is.

We now state that this corresponds to 68 and 95% confidence intervals.

- “ci” as shorthand for a confidence interval. Again, I couldn’t find this defined anywhere.

We believe this is standard notation, but for clarity we now specify this on the first occurrence in footnotes of abstract.